# Dereplication of Natural Extracts Diluted in Glycerin: Physical Suppression of Glycerin by Centrifugal Partition Chromatography Combined with Presaturation of Solvent Signals in ^13^C-Nuclear Magnetic Resonance Spectroscopy

**DOI:** 10.3390/molecules25215061

**Published:** 2020-10-31

**Authors:** Marine Canton, Jane Hubert, Stéphane Poigny, Richard Roe, Yves Brunel, Jean-Marc Nuzillard, Jean-Hugues Renault

**Affiliations:** 1Laboratoires Pierre Fabre Dermo-Cosmétique, 3 avenue Hubert Curien, BP 13562, CEDEX 1, 31035 Toulouse, France; marine.canton@univ-reims.fr (M.C.); spoigny@yahoo.com (S.P.); richard.roe@pierre-fabre.com (R.R.); yves.brunel@hotmail.com (Y.B.); 2Université de Reims Champagne Ardenne, CNRS, ICMR UMR 7312, CEDEX 2, 51097 Reims, France; jane.hubert@nat-explore.com

**Keywords:** natural extract, glycerin, chemical profiling, centrifugal partition chromatography, ^13^C nuclear magnetic resonance, cosmetic industry

## Abstract

For scientific, regulatory, and safety reasons, the chemical profile knowledge of natural extracts incorporated in commercial cosmetic formulations is of primary importance. Many extracts are produced or stabilized in glycerin, a practice which hampers their characterization. This article proposes a new methodology for the quick identification of metabolites present in natural extracts when diluted in glycerin. As an extension of a ^13^C nuclear magnetic resonance (NMR) based dereplication process, two complementary approaches are presented for the chemical profiling of natural extracts diluted in glycerin: A physical suppression by centrifugal partition chromatography (CPC) with the appropriate biphasic solvent system EtOAc/CH_3_CN/water 3:3:4 (*v*/*v*/*v*) for the crude extract fractionation, and a spectroscopic suppression by presaturation of ^13^C-NMR signals of glycerin applied to glycerin containing fractions. This innovative workflow was applied to a model mixture containing 23 natural metabolites. Dereplication by ^13^C-NMR was applied either on the dry model mixture or after dilution at 5% in glycerin, for comparison, resulting in the detection of 20 out of 23 compounds in the two model mixtures. Subsequently, a natural extract of *Cedrus atlantica* diluted in glycerin was characterized and resulted in the identification of 12 metabolites. The first annotations by ^13^C-NMR were confirmed by two-dimensional NMR and completed by LC-MS analyses for the annotation of five additional minor compounds. These results demonstrate that the application of physical suppression by CPC and presaturation of ^13^C-NMR solvent signals highly facilitates the quick chemical profiling of natural extracts diluted in glycerin.

## 1. Introduction

Natural extracts are commonly used all over the world as ingredients in cosmetic formulations for their antioxidant activity, their scent, their antimicrobial properties, and as natural dyes or hydration agents [1,2,3]. Over the last few years, regulation has become more demanding, as evidenced by the European cosmetics regulation in force since 2014 (no 1223/2009), which gives a prominent role to consumer safety [4,5]. This regulation includes a list of prohibited and restricted compounds. Moreover, it requires the suppliers to establish a product information file (PIF) for each cosmetic product. The PIF summarizes data including a description of the cosmetic product, the proof of the claimed effect, a safety assessment, and product stability data [6]. Consequently, the composition of the natural extracts contained in the cosmetic formulations must be known to prevent toxicity related risks.

The most common way to characterize an extract requires the isolation of its constituents [7]. Although being exhaustive, this process is very time consuming and usually leads to the purification of known compounds. Quicker methods more compatible with industry constraints must be found. To respond to this need, dereplication strategies mainly based on liquid chromatography hyphenated to mass spectrometry (LC-MS), liquid chromatography hyphenated to ultraviolet spectroscopy with diode array detection (LC-UV-DAD), thin layer chromatography (TLC), and/or nuclear magnetic resonance (NMR) data combined with the use of in silico or experimental databases were developed in order to avoid the purification of all components of a mixture [8,9,10]. The term dereplication has been introduced in the 1990s as a method for the quick identification of known compounds in a sample [11,12].

The dereplication process developed by Hubert et al. relies on ^13^C-NMR chemical shift comparisons for compound identification [10]. It starts with the extract fractionation by centrifugal partition chromatography (CPC) followed by ^13^C-NMR analysis of all fractions. After automatic collection and pooling of ^13^C-NMR signals, hierarchical clustering analysis (HCA) highlights clusters of ^13^C-NMR chemical shifts in which all elements share similar chromatographic behaviors. Finally, the clusters are assigned to chemical structures using a database that contains the chemical structures and calculated ^13^C-NMR chemical shifts of known natural metabolites of low molecular weight [10]. The structures that best match the experimental data are then confirmed by two-dimensional (2D) NMR and, if required, by complementary MS analysis. According to the Chemical Analysis Working Group of the Metabolomics Standard Initiative, this compound annotation process has a confidence level of 2 because it relies on spectral similarity [13,14]. After complementary techniques confidently defining 2D structures like 2D NMR, the confidence level increases to 1. The efficiency of this ^13^C-NMR based workflow has been largely proven by the identification of major compounds in numerous dry extracts [15,16,17,18].

Nowadays, many natural extracts of industrial interest are produced or stabilized in matrices or support solvents for easy handling and preservation [19,20,21]. Glycerin, also called glycerol, is the most used carrier solvent in marketed cosmetic ingredients, mainly due to its non-toxicity as well as its humectant and bacteriostatic properties [22]. In addition, it can be claimed as a natural ingredient when produced as a by-product of the synthesis of biodiesel [23]. A high concentration of glycerin in an extract interferes unfavorably with the dereplication procedure briefly described hereabove. The high boiling point of glycerin (BP = 286 °C at 105 Pa, with decomposition) and its very low vapor pressure (P_vap_ = 0,02 Pa at 25 °C) makes it impossible to evaporate without metabolite degradation [24].

The aim of this study was to develop an improved method for the quick identification of the major compounds contained in a glycerinated natural extract. Two strategies were jointly elaborated, including a physical suppression of glycerin by CPC and a spectroscopic suppression of the ^13^C-NMR resonances of glycerin. For the physical suppression of glycerin, an appropriate CPC biphasic system able to separate glycerin from other metabolites was investigated. Then, spectroscopic suppression by presaturation in ^13^C-NMR using a previously described method [25] was applied to fractions which still contained glycerin. As a proof of concept, the ^13^C-NMR dereplication methodology for glycerinated extracts was first applied to a model mixture of 23 analytical standards of diverse polarities. A comparison was performed between the metabolites identified in the dry model mixture and diluted at 5% in glycerin. Secondly, the method was tested on a genuine extract of *Cedrus atlantica* bark diluted in glycerin.

## 2. Results and Discussion

### 2.1. General Methodology for ^13^C-NMR Dereplication of Metabolite Mixtures Diluted in Glycerin

The aim of the present work was to develop a ^13^C-NMR dereplication process dedicated to natural extracts diluted in glycerin. As described by the flowchart in Figure 1, the approach consists of the combination of two glycerin removal methods: A physical suppression by CPC fractionation followed by a spectroscopic suppression of glycerin signals in ^13^C-NMR of glycerin-containing fractions.

CPC is a widely used tool for natural compound separation [26,27]. It is a solid-support-free chromatographic technique that uses the two liquid phases of a biphasic solvent system as stationary and mobile phases. The choice of the biphasic solvent system is of prime importance in this context. A high proportion of glycerin in a high volume of injected sample may disrupt the biphasic nature of the solvent system, resulting in separation failure [28]. Indeed, the two phases of the biphasic system must remain stable in the presence of a large amount of glycerin and, at the same time, must allow a gradual elution of metabolites during the fractionation process. To obtain a glycerin extract, the plant is usually treated with water or ethanol and, after filtration and evaporation of the ethanol when present, glycerin is added. In other cases, the plant is directly extracted in glycerin and water, and then the extract is filtered. Whatever the method used, the resulting extract composition is usually complex and contains metabolites covering a wide range of polarities. Our approach was to retain the glycerin in the stationary phase while eluting the compounds of interest. 

At the end of this first step, glycerin was expected to be contained in only a few CPC fractions. These fractions may contain sugars for which separation is not easily achieved. While the dry fractions can be directly analyzed by ^13^C-NMR, those containing glycerin in large proportions cannot. However, the global chemical profile of the natural extract requires the characterization of these fractions. The direct analysis by ^13^C-NMR of fractions containing predominantly glycerin can lead to the production of decoupling artifacts linked to the very high intensity of the two glycerin NMR signals [25]. A specific ^13^C signal suppression sequence was used to remove these artifacts and reduce the intensity of glycerin signals. A presaturation technique reported in [25] achieved a 97% signal intensity decrease, thus providing an efficient elimination of decoupling artifacts. Moreover, the chemical shift band for which a 50% signal intensity decrease occurred was sharp—about 0.1 ppm wide —avoiding information loss related to signal suppression.

The ^13^C-NMR spectra of glycerin-free and glycerin-containing fractions were subjected to the usual dereplication methodology: Automatic collection and alignment of ^13^C peaks, HCA and identification of metabolites present in a database by ^13^C chemical shift fingerprint comparison. The feasibility of this strategy was demonstrated on a model mixture containing 23 analytical standards from different chemical families as well as by the characterization of a genuine glycerin extract of *Cedrus atlantica* Carrière.

### 2.2. Proof of Concept on a Model Mixture

To simulate the chemical diversity of natural extracts obtained with non-selective solvents such as water, alcohol, or glycerin, 23 analytical standards of various phytochemical classes have been selected. Sugars, organic acids, fatty acids, phenolic acids, hydroxycinnamic acids, stilbenes, tannins, flavonoids, alkaloids, saponins, and amino acids were part of the model mixture. In the case of flavonoids and hydroxycinnamic acids, the relatively lipophilic aglycone form (quercetin and ferulic acid), as well as the glycosylated form (rutin and chlorogenic acid) were included. For comparison purposes, two model mixtures were prepared: One in dry form and one at 5% wt. in glycerin/water (1:1 *w*/*w*). 

The first step of our method was to fractionate the model mixtures by CPC with an appropriate solvent system. The biphasic system ethyl acetate (EtOAc)/acetonitrile (CH_3_CN)/water 3:3:4 (*v*/*v*/*v*) was investigated because of the combination of a medium polar binary system (EtOAc/water) with CH_3_CN as a bridging solvent was promising for our study. This type of system has already been referenced for the fractionation of natural polar extracts [29,30,31]. The influence of glycerin on the biphasic system EtOAc/CH_3_CN/water 3:3:4 (*v*/*v*/*v*) was evaluated according to the method developed by Marchal et al. [32], which relies on the drawing of a pseudo ternary diagram in which the mobile phase, the stationary phase, and glycerin intervene at the apexes. Glycerin was successively added to mobile and stationary phases present in predefined ratios: 0/1, 1/3, 1/1, 3/1, and 1/0 (*w*/*w*). Then, the solvent mixture was shaken, and the visual inspection for homogeneity of one or two phases was carried out. If the mixture becomes monophasic, the corresponding composition (*w*/*w*/*w*) of this mixture in terms of upper phase, mobile phase, and glycerin was reported on the phase diagram in its orthogonal representation, and then the binodal curve was drawn by connecting the different demixing points. 

The biphasic nature of the EtOAc/CH_3_CN/water 3:3:4 (*v*/*v*/*v*) system was very little modified by the addition of glycerin (maximum tested mass of glycerin was 90% wt. without forming a single phase), which promises high stability of this solvent system (Figure 2a). Conversely, the same experiment carried out on *n*-butanol (*n*-BuOH)/acetic acid/water 4:1:5 (*v*/*v*/*v*), one of the most polar CPC systems, resulted in single-phase solvent mixtures. The maximum demixing point was reached at 47% wt. of glycerin added in the 1:1 (*w*/*w*) lower/upper phase mixture. The pseudo ternary system of glycerin in *n*-BuOH/acetic acid/water 4:1:5 (*v*/*v*/*v*) is presented in Figure 2b as an example of a CPC solvent system which is not suitable for glycerin-containing samples.

The selection of the biphasic solvent system EtOAc/CH_3_CN/water 3:3:4 (*v*/*v*/*v*) as a solvent system compatible with the presence of high amounts of glycerin was followed by an evaluation of its suitability for natural product fractionation. The partition coefficient (K_D_) indicates the distribution of a compound in the upper and the lower phases of a biphasic system. For a crude plant extract containing compounds covering a wide range of polarities, the ideal CPC system for our study should cover a large range of K_D_ values with a significantly different one for glycerin. Moreover, the partition coefficients K_D_ of the compounds of interest should be comprised between 0.2 and 5 to retain the highest selectivity while maintaining short CPC run times [33]. The partition coefficients of glycerin and ten metabolites from various chemical families and with diverse polarities contained in the model extract were measured in the EtOAc/CH_3_CN/water 3:3:4 (*v*/*v*/*v*) solvent system: Caffeine, chlorogenic acid, ferulic acid, glycyrrhizin, linoleic acid, polydatin, quercetin, rutin, succinic acid, and vanillin. The shake flask method [34] was used to determine each K_D_ individually. The construction of response-concentration curves was performed for each of the ten compounds, ensuring that each analytical measurement was carried out within the linearity range (Appendix A). The measured partition coefficients of the ten standards, shown in Table 1, ranged from 0.02 for the low polarity linoleic acid to 5.28 for rutin, a medium-polar compound. This result showed a large K_D_ diversity, thus predicting a successful separation of metabolites by CPC elution. Moreover, we measured the K_D_ of glycerin using the shake flask method and an analysis by gas chromatography with flame ionization detection GC/FID (data not shown). The glycerin did not distribute at all in the system and remained exclusively present in the lower phase. CPC in the ascending mode was thus selected for the elution of standard compounds in order to maintain glycerin in the column. The system EtOAc/CH_3_CN/water 3:3:4 (*v*/*v*/*v*) seemed appropriate for the resolution of our initial problem: It is stable in the presence of glycerin and provides a good selectivity over a wide range of analyte polarities. The obtained result provides a way to physically separate glycerin from the other compounds and thus obtain, after CPC fractionation, glycerin-free fractions from the mobile phase and glycerin-containing fractions from the stationary phase, as collected by column extrusion (i.e., back-flushing of the column stationary phase by fresh stationary phase).

The CPC fractionation of the two model mixtures containing 23 analytical standards in a dry form or diluted at 5% wt. in glycerin/water 1:1 (*w*/*w*) was performed. The injected sample represented 2 g of the dry model mixture, which corresponded to 40 g of the glycerin-containing model mixture. The injected sample volume was approximately 35 mL or 12% of the column capacity, which is still acceptable and sufficient for the compounds to be partitioned in the remaining part of the column. No disturbance of the solvent system was observed in the fractionation of the glycerin-containing mixture. As a result, 13 simplified fractions were obtained in order of decreasing polarity of their constituents. Sample recoveries were 92% and 110% wt. for the dry and glycerinated model mixtures, respectively. The hygroscopic character of glycerin explains the recovery of more than 100%. The details of the CPC fractionation mass balances are presented in Appendix A. Regarding the elution fractions (F_01_–F_09_ in each experiment), the recovered masses were almost similar in both experiments, 311 mg and 320 mg for dry and glycerin extracts, respectively. Fractions F_10_ to F_13_ correspond to the column extrusion step (see experimental part). The summary HPTLC chromatograms allowed a visual analysis of the two CPC fractionations in Figure 3.

According to the high-performance (HP)TLC plates of the analytical standards available in Appendix A, several standards have been assigned: 1: Linoleic acid; 2: Quercetin; 3: Ferulic acid, vanillin; 4: Gallic acid; 5: Polydatin; 6: Glycyrrhizin; 7: Chlorogenic acid; 8: Rutin; 9: Glycerin; 10: d-fructose, d-glucose, and sucrose.

As shown in Figure 3a, the blue polar spot of glycerin was only present in fractions F_11_, F_12_, and to a lesser extend in F_13_, confirming the non-partitioning of glycerin in the CPC biphasic system. Moreover, the presence of glycerin was easily deduced because the drying of these fractions left a liquid residue. These three fractions corresponded to the extrusion of the CPC column. Glycerin was maintained in the stationary phase, while many analytical standards could be eluted during the CPC fractionation. 

As expected from partition coefficient measurements, the analytical standards were relatively well separated, which was highly encouraging for future dereplication works. Fractionation of the dry sample led to a separation almost identical to that obtained with the glycerinated mixture. Some analytical standards were eluted in exactly the same fractions, while the elution of others was slightly modified. For example, gallic acid and polydatin (blue spot at a retardation factor (Rf) of 0.54 and 0.21 in Figure 3b,d, respectively) were both predominantly present in fractions F_03_ and F_04_. Chlorogenic acid (Rf 0.11 blue spot in Figure 3b,d) was eluted in fractions F_05_–F_07_ and F_05_–F_08_ for glycerinated and dry mixture fractionation, respectively. Sugars (browns spot at Rf 0.03 in Figure 3a,c), as very polar compounds, were present in the three last fractions in both experiments. Some differences appeared for quercetin (yellow spot at Rf 0.63 in Figure 3b,d), which eluted more slowly in the fractionation of the glycerinated mixture than for the dry mixture. These differences can be explained by the interface and/or flow pattern modifications due to the larger volume of glycerin-rich injected samples.

Glycerin-free fractions were directly characterized by ^13^C-NMR spectroscopy. The three glycerin-containing fractions F_11_, F_12_, and F_13_ were analyzed by ^13^C-NMR with presaturation of glycerin signals. 200 mg of fractions were mixed with 600 μL of DMSO-d_6_. Such a quantity was necessary for metabolite detection. Figure 4 presents the results of the presaturation of the two signals of glycerin in fraction F_11_. 

Without presaturation (Figure 4a), the resonance peaks of DMSO-*d*_6_ were much smaller than those of glycerin. After presaturation (Figure 4b), glycerin signals were significantly reduced, thus demonstrating the efficiency of the glycerin resonance peak elimination. Quantitatively, there was a 97% decrease in the signal at 63.7 ppm and a decrease of 94% for the signal at 73.1 ppm. Furthermore, decoupling artifact signals were present in the spectrum without presaturation (Figure 4a). Some of these were easily recognizable because they were out of phase, but others were easy to confuse with genuine signals in the spectrum. These decoupling artifacts could interfere with the dereplication process by presuming signals where there were none. Presaturation of glycerin signals also led to the removal of decoupling artifacts, as expected.

The alignment of the ^13^C-NMR data of the 13 fractions could then be carried out. The 2D heat map visualization and the HCA were performed. Figure 5 shows the heat maps obtained for the glycerinated model mixture after HCA on the 13 fractions with a classical ^13^C-NMR analysis (Figure 5a,b) and with glycerin signals presaturation on fractions F_11_, F_12_, and F_13_ (Figure 5c). 

Figure 5a seemed to have mainly one cluster, as indicated by the large correlation branch of the HCA made on chemical shift lines. The manual suppression of this intense cluster (δ 63.7 and 73.1) in the text file before visualization in 2D heat map improved chemical shift clusters observation (Figure 5b). The presaturation of glycerin signals in the fractions F_11_, F_12_, and F_13_ further improved the heat map with well-defined chemical shift clusters (Figure 5c) and fewer glycerin artifacts. In addition to the decoupling artifacts, other parasitic signals related to the truncation of the intense glycerin signals were observed, as highlighted in Appendix A. The presaturation allowed a strong decrease in the intensity of the glycerin signals and the strong reduction of all the artifacts related to glycerin.

The chemical profiling of model mixtures was performed by database search (see experimental part). Heat maps and annotations of chemical shift clusters are given in Figure 6. The same 20 analytical standards were characterized in the dry and the glycerinated mixtures.

The most apolar compounds, linoleic acid, ferulic acid, vanillin, and quercetin, were the first to elute (clusters 1, 6, 7, and 9, respectively). This result was expected given their K_D_, which was less than or equal to 0.1 in this solvent system. Polydatin (cluster 8), which has a K_D_ around 0.7, was eluted from fraction three in both experiments. Caffeine, chlorogenic acid, and succinic acid (clusters 5, 10, 13 respectively), which have K_D_ between 1 and 1.5, were then eluted. Rutin, with the highest K_D_ (5.3), emerged from the column at the end of the process. Only the elution behavior of glycyrrhizin was not in agreement with its measured K_D_, which could be due to its strong amphiphilic character. The simultaneous measurement of the K_D_ of the ten metabolites in the model mixture was undertaken (Table 1). Among the ten compounds, glycyrrhizin and rutin presented a strongly changed K_D_ value, depending on measurement condition, either as a pure compound or as a mixture component. The K_D_ value of rutin was high individually and in a mixture, which explained its late elution. However, for glycyrrhizin, the K_D_ measured when pure was greater than 2 whereas it was less than 1 when in a mixture. This result explained the behavior of glycyrrhizin, which eluted starting from the third fraction.

Among the 23 metabolites constituting the model mixture, three metabolites could not be annotated: Tannic acid, citric acid, and proline. With hindsight, tannic acid was found not to be an appropriate candidate for this study. Commercial tannic acid was usually a gallotanins mixture which tended to undergo autodegradation [35,36]. We thus chose to exclude it from the discussions. Regarding proline and citric acid, two hypotheses were put forward. A long lasting elution of a compound could generate a very low concentration in each of the fractions and then a non-observation of the signals in ^13^C-NMR. A second explanation, the most likely, was that the clusters of the two missing compounds were broken up. Indeed, when a compound has one or more chemical displacements very close to another compound, the usual δ tolerance for cluster detection is unusable. For example, several chemical shifts of proline can be in the same 0.2 ppm bin as polydatin (δ 62.1), glycyrrhizin (δ 46.8), or caffeine (δ 30.0). Additional experiments were performed on each CPC fraction by LC-MS in order to observe the elution of the analytical standards. Figure 7 shows the elution profile of the two missing compounds: Citric acid and proline. The graphical visualization of their elution showed a similar elution behavior in dry and glycerin extracts fractionation. Moreover, their elution was limited to a few fractions. The hypothesis of a broken cluster on the heat map was therefore preferred.

These results confirmed that despite the large amount injected and the high proportion of glycerin in the extract, it was possible to characterize the compounds contained in a model mixture diluted in glycerin with the method described hereafter: A physical suppression of glycerin by CPC fractionation with the EtOAc/CH_3_CN/water 3:3:4 (*v*/*v*/*v*) solvent system in ascending mode followed by a spectroscopic suppression of glycerin signals for fractions containing glycerin in large proportion. In addition, the proof of concept of this method for the quick chemical profiling of glycerin extracts was promising since the results of the dereplication of the dry and of the glycerinated form of the same mixture were identical. It was then necessary to demonstrate the robustness of this method by the analysis of a genuine glycerinated extract.

### 2.3. Dereplication of a Genuine Natural Extract Diluted in Glycerin

The selected extract was an experimental aqueous extract of *Cedrus atlantica* Carrière bark diluted in glycerin. Due to its origin, this aqueous extract contained a complex mixture of metabolites covering a wide polarity range. The major compounds were identified by the glycerin-insensitive new dereplication method illustrated by the flow chart in Figure 1.

CPC fractionation was carried out as described for the model mixture. The injection of 17.5 g of glycerinated extract was equivalent to the injection of 2.5 g of dry extract as the solution contains 14.5% wt. of dry natural extract. The fractionation process resulted in 13 fractions. Eleven of them could be evaporated to dryness whilst the last two fractions contained glycerin. The recovery of the sample was 99%. The mass of each CPC fraction is available in Appendix A. The first nine fractions and the last two extruded fractions represented 1% wt. and 98.5% wt. of the recovered material, respectively. As shown on the HPTLC profiles (Appendix A), the extruded fractions contained glycerin and sugars predominantly. 

Spectroscopic glycerin suppression was therefore undertaken by ^13^C-NMR presaturation on F_12_ and F_13_ in order to decrease glycerin signal intensity and to remove possible decoupling artifacts. ^13^C-NMR spectra were then processed with an alignment of NMR data in regular chemical shift windows of 0.3 ppm width. Indeed, even though glycerin was removed by spectroscopy, significant matrix effects appeared in the last two fractions. Table 2 reports these matrix effects on the chemical shifts at the anomeric positions of the five sugars present in fractions F_12_ and F_13_.

Chemical shift bins, of widths generally set at 0.2 ppm to be selective enough while including a possible chemical shift deviation, therefore had to be increased to 0.3 ppm. HCA was then performed. The resulting heat map containing the correlated ^13^C-NMR signal revealed ten major chemical shift groups (Figure 8). The assignment of the clusters led to the identification of 12 metabolites: *p*-hydroxybenzoic acid, protocatechuic acid, quinic acid, *p*-coumaric acid, shikimic acid, 4-hydroxybenzoic acid 4-*O*-glucoside, pyroglutamic acid, α-(D)-glucose, β-(D)-glucose, α-(D)-fructofuranose, β-(D)-fructopyranose, β-(D)-fructofuranose. Clusters one and two referred to residual signals from glycerin and a glycerin derivative identified as glycerin acetate.

The structures of all the compounds annotated on the heat map were confirmed by checking for all carbon atoms of the neighboring correlations in HSQC, HMBC, and COSY spectra. Other metabolites reported in the genus *Cedrus* [37] were finally searched by complementary LC-MS analyses, leading to the annotation of catechin, taxifolin, and astragalin in the less polar fractions and syningetin-3-*O*-glucoside and kaempferol-3-*O*-rutinoside in the medium polar fractions. The work presents a ^13^C-NMR dereplication process permitting the characterization of major compounds of a natural extract diluted in glycerin. LC-MS, as a complementary technique with very high sensitivity, allows annotating other minor compounds referenced in the literature.

## 3. Materials and Methods 

### 3.1. Chemical and Natural Extract

Ethyl acetate, acetonitrile, *n*-butanol, and methanol (MeOH) were purchased from Carlo Erba (Val de Reuil, France). Acetic acid was purchased from VWR (Radnor, PA, USA). Two model mixtures were prepared with 23 analytical standards for a total mass of 2 g: 500 mg of sucrose, 500 mg of D-glucose, 500 mg of D-fructose, and 25 mg of each of the following 20 compounds: Caffeine, chlorogenic acid, choline, citric acid, ferulic acid, gallic acid, glycerin, glycyrrhizin, linoleic acid, malic acid, polydatin, L-proline, quercetin, rutin, D-sorbitol, succinic acid, tannic acid, uracil, vanillin, and xylitol. The purity of each analytical standard was at least 95%. Their suppliers are listed in Appendix A. While one mixture was left as the dry model sample, the second was diluted at 5% wt. in glycerin and water in a ratio of 1:1 (*w*/*w*), giving a model mixture diluted in glycerin (glycerinated model mixture, hereafter) of 40 g. The genuine natural extract was a glycerinated extract of *Cedrus atlantica* Carrière. It was produced by maceration of *C. atlantica* bark in water, followed by clarification, concentration, and glycerin addition, resulting in a 14.5% wt. *C. atlantica* extract in 75.5% wt. glycerin and 10% wt. water.

### 3.2. Distribution Coefficients Determination

The distribution coefficients of glycerin and of ten analytical standards (caffeine, chlorogenic acid, ferulic acid, glycyrrhizin, linoleic acid, polydatin, quercetin, rutin, succinic acid, and vanillin) were measured individually or as a mixture in the EtOAc/CH_3_CN/water 3:3:4 (*v*/*v*/*v*) biphasic system. The shake flask method was used to determine each K_D_ [34]. The temperature of the study was 20 °C. After solvent system equilibration, a defined volume of upper and lower phase (5 mL for glycerin study, 2 mL for analytical standards study) were transferred into a vial and vortexed for 5 s in the presence of the compound (11 mg for glycerin, 3 mg for the other analytical standards). Each vial was then placed for 15 min in an ultrasonic bath and left to settle (centrifugation was required in the case of glycyrrhizin to break the emulsion and separate the two phases). 500 µL of the lower phase and 500 µL of the upper phase were separately transferred into two clean vials. The solvents were evaporated in a vacuum oven at 40 °C. Finally, the lower and upper phase residues were diluted in 1 mL of MeOH for LC-MS analysis. The distribution coefficient K_D_ is defined as the ratio of the concentration of the solute in the upper phase (C_up_, organic phase) to its concentration in the lower phase (C_down_, aqueous phase) at thermodynamic equilibrium.
(1)$$K_{D}=\frac{C_{\mathrm{up}}}{C_{\mathrm{down}}}$$

For K_D_ measurement in the mixture, a total of 240 mg of the 23 analytical standards in the same proportion as in the model sample were diluted in 160 mL of each lower and upper phase of the solvent system. After ultrasonic agitation for 5 min, the two phases were separated, evaporated under vacuum, and the residues were diluted in 3 mL MeOH for LC-MS analyses. 

Measurements were performed in triplicate. Means of the LC-MS peak areas and standard deviations were calculated to give a distribution coefficient value (Table 1).

### 3.3. LC-MS Analyses

LC-MS analyses were performed with an Acquity UPLC H-Class (Waters, Manchester, UK) system coupled to a Synapt G2-Si (Waters) equipped with an electrospray (ESI) ion source. Chromatographic separation was achieved on a Kinetex C18 column (150 × 2.1 mm, 2.6 μm; Phenomenex, Torrance, CA, USA). The column temperature was regulated at 30 °C. Compounds were eluted with a gradient of water (Eluent A) and CH_3_CN (Eluent B) with 0.1% formic acid in each mobile phase. The mobile phase flow rate was maintained at 0.6 mL/min and the gradient was designed as follows: t = 0 min, 5% B; t = 1 min, 5% B; t = 9 min, 40% B; t = 15 min, 100% B; t = 17 min, 100% B; t = 17.1 min, and 5% B until 19 min. All injection volumes were 0.5 μL. MS analyses were performed in both positive (ESI^+^) and negative (ESI^−^) ion modes. For each compound, the most suitable mode was chosen. Electrospray interface fitted to the following parameters: Capillary voltage 3 kV and 2 kV for ESI^+^ and ESI^−^, respectively; desolvation temperature 450 °C; desolvation gas flow 950 L/h; source temperature 120 °C; cone voltage 40 V; cone gas flow 50 L/h and scanning range of m/z 50–2000 both in the positive and negative ionization modes.

### 3.4. Centrifugal Partition Chromatography

CPC experiments were carried out with a lab-scale FCPE300^®^ column of 303.5 mL capacity (Kromaton Technology, Angers, France). The column was composed of seven circular partition disks. Each disk is engraved with 33 twin-cells. The liquid phases were pumped by a preparative 1800 V7115 pump (Knauer, Berlin, Germany). All experiments were performed in the same conditions. The biphasic system EtOAc/CH_3_CN/water 3:3:4 (*v*/*v*/*v*) was prepared in a separatory funnel. After decantation, the phases were separated. CPC fractionation was achieved in the ascending mode: The lower phase was used as the stationary phase and the upper one as the mobile phase. The CPC column was filled at a flow rate of 100 mL/min and at a rotation speed of 500 rpm. Then the rotation speed was increased to 1200 rpm. For the dry extract, 2 g of dry extract was dissolved in 12 mL of stationary phase and 6 mL of mobile phase and injected through a 20 mL injection loop. The glycerinated mixtures were directly injected through a 35 mL injection loop (40 g of the model glycerinated mixture and 17.5 g of the *C. atlantica* extract, respectively). The mobile phase was pumped progressively from 0 to 20 mL/min in 5 min and then maintained at 20 mL/min for 60 min. The most hydrophilic compounds retained inside the column were finally extruded by pumping fresh stationary phase for 25 min, maintaining the flow rate at 20 mL/min. Fractions of 20 mL were collected by a Labocol Vario 4000 (Labomatic Instruments, Allschwil, Switzerland). All collected fractions were spotted on Merck TLC plates coated with silica gel 60 F_254_ and developed with EtOAc/toluene/acetic acid/formic acid (6:4:1:1 (*v*/*v*/*v*/*v*) for the model mixtures and 7:3:1:1 (*v*/*v*/*v*/*v*) for the genuine extract). After inspection at 254 nm, 366 nm, and under visible light after chemical revelation with acidified vanillin solution, fractions were pooled according to their composition. The pooling after fractionation of the two model mixtures was identical and resulted in 13 elution fractions. The genuine natural extract was also fractionated in 13 fractions. All fractions were dried under vacuum, except several extruded fractions still containing glycerin.

### 3.5. Nuclear Magnetic Resonance

NMR analyses were performed at 298 K on an Avance AVIII-600 spectrometer (Bruker, Karlsruhe, Germany) equipped with a cryoprobe optimized for ^1^H detection and with cooled ^1^H, ^13^C and ^2^H coils and preamplifiers. Dry fractions were dissolved in 600 µL of DMSO-*d*_6_ and analyzed with the Uniform Driven Equilibrium Fourier Transform (UDEFT) sequence with an acquisition time of 0.36 s, and a relaxation delay of 3 s. 1024 scans were recorded on samples containing more than 20 mg, 2048 scans on those weighing between 10 mg and 20 mg and 4096 scans for fractions weighing less than 10 mg in order to obtain comparable signal-to-noise ratios. The receiver gain was set to the highest possible value. The spectra were manually phased and baseline-corrected using the TOPSPIN v4.0.5 software (Bruker). The central resonance of DMSO-*d*_6_ was set at 39.8 ppm for spectrum referencing. Additional homonuclear correlation spectroscopy (COSY), heteronuclear single-quantum coherence (HSQC), and heteronuclear multiple bond correlation (HMBC) spectra were also recorded on dry fractions. For fractions in which glycerin was still present in a large proportion, the NMR sample was prepared with 200 mg of the fraction in 600 µL of DMSO-*d*_6_. The presaturation sequence described in Canton et al. [25] was used to decrease the two dominating ^13^C-NMR signals of glycerin and facilitate the identification of metabolites eluted in these fractions. The presaturation field of intensity Ω_1_/2π = 11.7 Hz was focused on the two resonances of glycerin applied during 3 s before the 90° pulse in a modified Bruker *zgpg* pulse sequence. 

### 3.6. Data Processing

All ^13^C-NMR signals were automatically collected on each spectrum. The resulting peak lists were stored in a text file from which a locally developed algorithm written in Python language aligns the NMR peak positions in regularly spaced chemical shift windows (Δδ = 0.2 ppm for the model studies and 0.3 ppm for the genuine extract study). The table of peak intensity values according to chemical shift bin and fraction indexes was imported into PermutMatrix software (version 1.9.3, LIRMM, France) for hierarchical clustering analysis. The resulting ^13^C-NMR chemical shift clusters were visualized as dendrograms on a 2D heat map. In order to identify compounds, the elements of each chemical shift cluster were used as a search key in a local database built using the ACD/C+H NMR Predictors and DB software (Advanced Chemistry Development, Inc., ACD/Labs, Toronto, ON, Canada). This database contains more than 3000 compounds to date and associates structures to the predicted NMR chemical shifts of the proton and carbon atoms, as calculated by the ACD/Labs predictor. A literature survey was carried out on the genus *Cedrus*, resulting in 40 metabolites stored in the database [37,38,39,40]. A tolerated ^13^C-NMR chemical shift difference between the predicted database spectrum and the real spectrum was established at 2 ppm. Finally, each proposition given by the database was confirmed by the interpretation of 1D and 2D NMR data (^1^H NMR, HSQC, HMBC, and COSY) and LC-MS analyses.

### 3.7. High-Performance Thin-Layer Chromatography

HPTLC analyses were performed on Merck HPTLC plates 10 × 20 coated with silica gel 60 F_254_ with an automatic TLC sampler (ATS 4) and an automatic development chamber (ADC 2) (CAMAG, Muttenz, Switzerland). 5 μg of pure analytical standards, 12 μg of dry CPC fractions or 80 μg of glycerin-containing CPC fractions were applied with a band length of 8 mm. The developing solvents were: EtOAc/toluene/acetic acid/formic acid 6:4:1:1 (*v*/*v*/*v*/*v*) for the analysis of standards and of the CPC fractions obtained from the model mixtures while the same developing solvents in 7:3:1:1 (*v*/*v*/*v*/*v*) proportions were used for the *C. atlantica* extract. First, the plates were visualized at 254 nm and 366 nm. Then, the plates were heated at 100 °C with a TLC plate heater 3 (CAMAG) for 3 min and immersed in a solution of Neu reagent (1.25 g of diphenylborinic acid aminoethylester dissolved in 250 mL of EtOAc) and recorded at 366 nm and in white light. Finally, the plates were immersed in an acidified vanillin solution (375 mg of vanillin, 3.75 mL of 96% H_2_S0_4_ diluted in 125 mL of ethanol) and heated at 100 °C for 5 min. An HPTLC chromatogram of the 23 analytical standards, available in Appendix A, was performed to calculate their retardation factor. HPTLC analyses were carried out at the end of each CPC fractionation.

## 4. Conclusions

The combination of physical suppression by CPC fractionation and spectroscopic suppression by presaturation in ^13^C-NMR were reported for the dereplication of natural extract metabolites diluted in glycerin. The biphasic solvent system EtOAc/CH_3_CN/water 3:3:4 (*v*/*v*/*v*) was selected for its stability in the presence of glycerin and its ability to separate compounds in a wide polarity range. The proof of principle of this approach was successfully achieved on a model mixture of 23 analytical standards. The 20 compounds identified in the dry model mixture were also identified in the glycerinated one. The method applied on a bark extract of *Cedrus atlantica* Carrière diluted in glycerin resulted in the identification of 12 metabolites of different chemical classes. This approach is, therefore, efficient for the fast chemical profiling of natural extracts diluted in glycerin. In the future, the characterization of major metabolites in extracts diluted in other solvents, notably glycols, produced for the cosmetics market, will be studied.

## Figures and Tables

**Figure 1 molecules-25-05061-f001:**
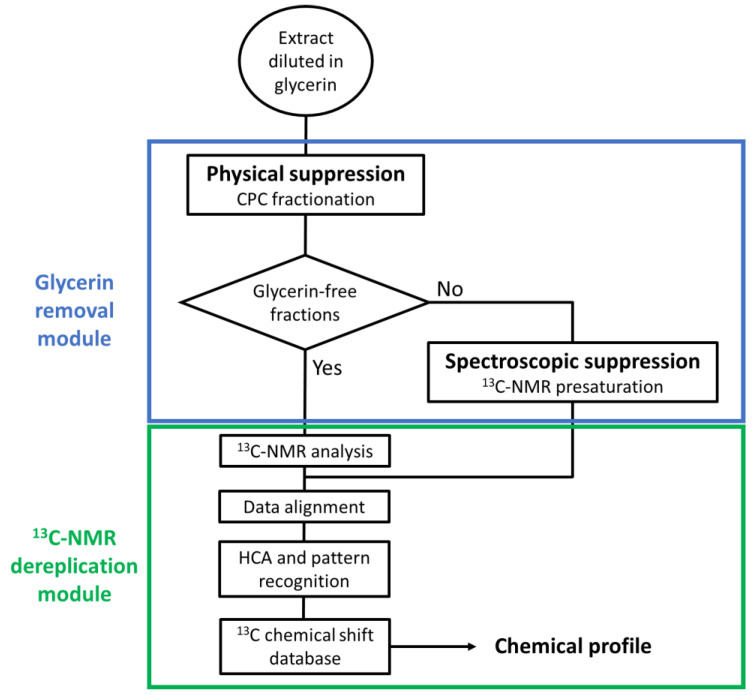
Global workflow for the dereplication of natural extracts diluted in glycerin. CPC: Centrifugal partition chromatography; HCA: Hierarchical clustering analysis.

**Figure 2 molecules-25-05061-f002:**
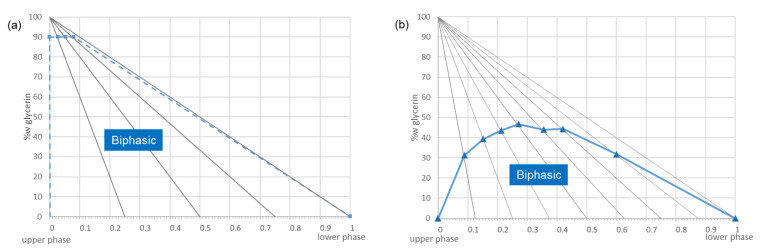
Ternary diagram of upper phase, lower phase, and glycerin (*w*/*w*/*w*) for the solvent systems: (**a**) ethyl acetate (EtOAc)/acetonitrile (CH_3_CN)/water 3:3:4 (*v*/*v*/*v*) and (**b**) *n*-butanol (n-BuOH)/acetic acid/water 4:1:5 (*v*/*v*/*v*) as an example of appropriate (**a**) and inappropriate (**b**) solvent systems for our physical glycerin suppression strategy. Demixing points are symbolized by triangles. In (**a**), no demixing point was observed until the addition of 90% wt. of glycerin (maximum points studied symbolized by squares). Experiments were performed at 20 °C.

**Figure 3 molecules-25-05061-f003:**
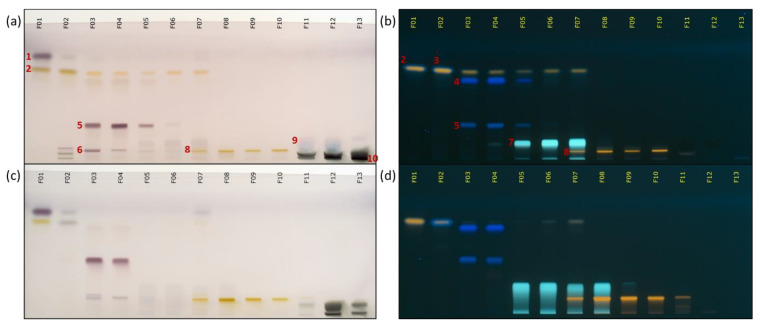
Recapitulative high performance (HP)TLC plates of the CPC fractionations. (**a**,**b**) refer to glycerin model mixture; (**c**,**d**) refer to dry model mixture. Vanillin/sulphuric acid reagent was used for plates (**a**,**c**) while (**b**,**d**) were revealed by the Neu reagent.

**Figure 4 molecules-25-05061-f004:**
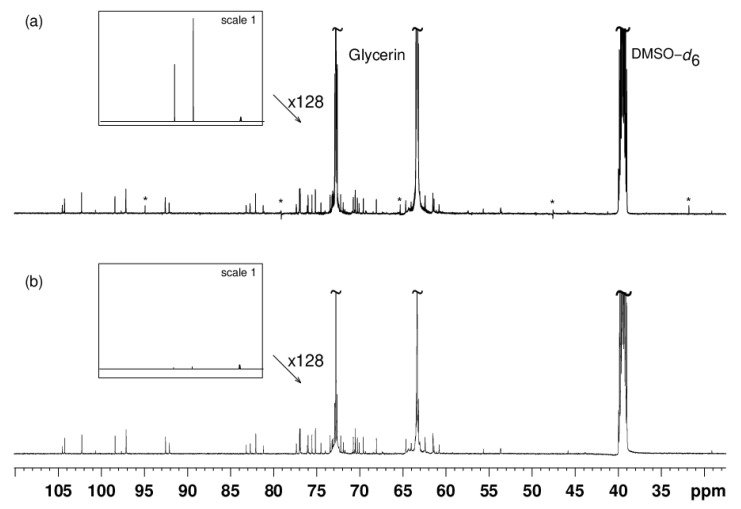
(**a**) ^13^C-NMR spectrum of glycerin-containing fraction F_11_. * refers to decoupling artifacts. (**b**) Analysis of the same sample as in (**a**) with multiple presaturation of glycerin signals. The frame inserts show spectra overviews drawn at the same vertical scale, without peak clipping. Both acquisitions required the recording of 1024 scans and eight dummy scans.

**Figure 5 molecules-25-05061-f005:**
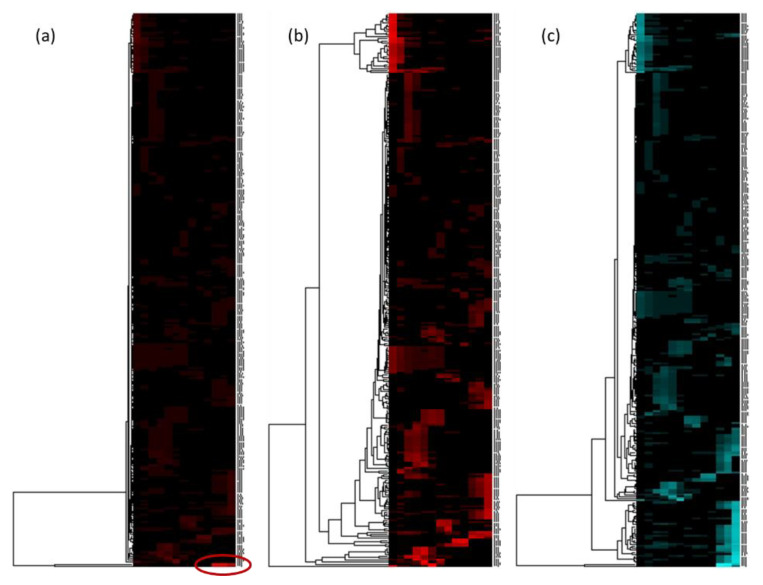
Comparison of heat maps for glycerin model extract: (**a**) without presaturation of glycerin signals. The red circle corresponds to the intense cluster of glycerin (δ 63.7, 73.1); (**b**) similar to (**a**) with manual suppression in the text file of the two bins 63.7 and 73.1 ppm corresponding to the glycerin signals; (**c**) with presaturation of glycerin signals during ^13^C-NMR analysis of F_11_, F_12_, and F_13_.

**Figure 6 molecules-25-05061-f006:**
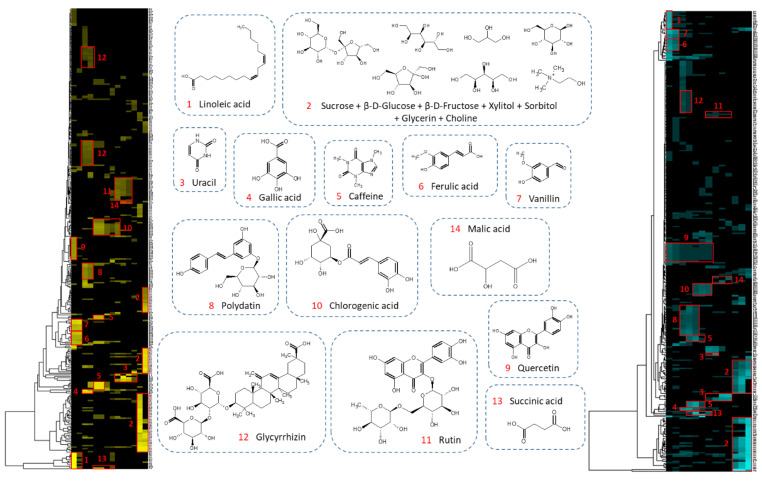
^13^C-NMR chemical shift clusters obtained by applying HCA on dry model extract (left) and diluted in glycerin (right).

**Figure 7 molecules-25-05061-f007:**
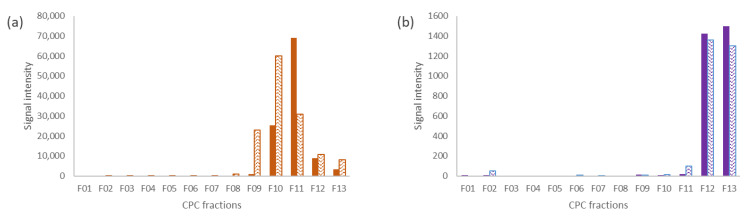
LC-MS elution profiles of (**a**) citric acid and (**b**) proline in dry and glycerin model mixture CPC fractionations. The full boxes correspond to the area of the signal in the fractionation of the dry extract and the hatched boxes to the fractionation of the glycerin extract.

**Figure 8 molecules-25-05061-f008:**
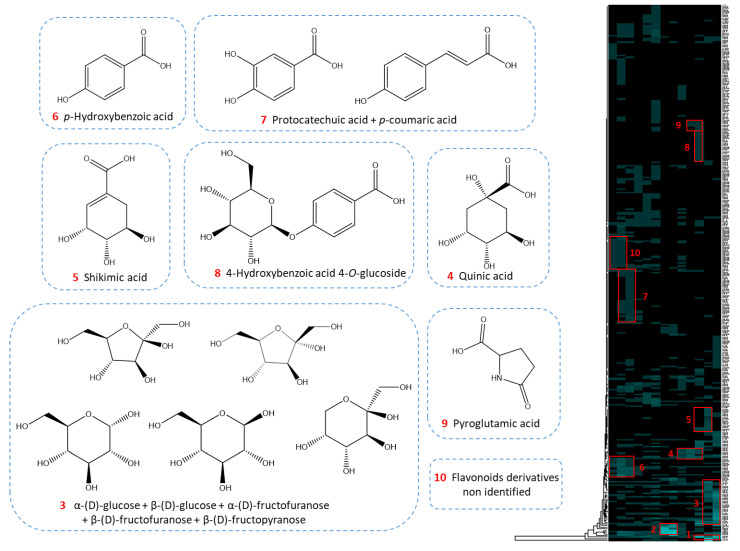
^13^C-NMR chemical shift clusters obtained by applying HCA on CPC fractions of glycerinated extract of *C. atlantica*.

**Table 1 molecules-25-05061-t001:** Analytical standards selected for the measurement of their distribution coefficients (K_D_) individually and as a mixture in the biphasic system EtOAc/CH_3_CN/water 3:3:4 (*v*/*v*/*v*). Their thin-layer chromatography (TLC) retardation factor (Rf) with EtOAc/toluene/acetic acid/formic acid 6:4:1:1 (*v*/*v*/*v*/*v*) as elution solvents, retention time and scan mode chosen are listed. Each LC-MS analysis was performed in triplicate. ESI: Electrospray ion.

Compound	Chemical Family	Formula	Rf	Retention Time (min)	Exact Mass (g/mol)	Scan Mode	Individual K_D_ by LC-MS	K_D_ in Mixture
Caffeine	Alkaloid	C_8_H_10_N_4_O_2_	0.39	3.63	194.0804	ESI^+^	1.13 ± 0.09	1.03 ± 0.01
Chlorogenic acid	Hydroxycinnamic acid	C_16_H_18_O_9_	0.11	3.69	354.0951	ESI^−^	1.11 ± 0.09	1.23 ± 0.01
Ferulic acid	Hydroxycinnamic acid	C_10_H_10_O_4_	0.66	5.42	194.0579	ESI^−^	0.14 ± 0.01	0.17 ± 0.01
Glycyrrhizin	Saponin	C_42_H_62_O_16_	0.04	10.15	822.4038	ESI^−^	2.09 ± 0.13	0.78 ± 0.02
Linoleic acid	Fatty acid	C_18_H_32_O_2_	0.81	14.67	280.2402	ESI^−^	0.02 ± 0.01	0.01 ± 0.01
Polydatin	Stilbene	C_20_H_22_O_8_	0.21	5.55	390.1315	ESI^−^	0.62 ± 0.02	0.72 ± 0.01
Quercetin	Flavonoid	C_15_H_10_O_7_	0.63	7.81	302.0427	ESI^+^	0.02 ± 0.01	0.05 ± 0.01
Rutin	Flavonoid	C_27_H_30_O_16_	0.04	5.61	610.1534	ESI^+^	5.28 ± 0.34	2.52 ± 0.14
Succinic acid	Organic acid	C_4_H_6_O_4_	-	0.80	118.0266	ESI^−^	1.18 ± 0.09	1.24 ± 0.14
Vanillin	Other phenolic	C_8_H_8_O_3_	0.66	4.55	152.0473	ESI^+^	0.08 ± 0.01	0.17 ± 0.01

**Table 2 molecules-25-05061-t002:** Matrix effects of chemical shifts of sugars anomeric position.

Compound	δ in F_12_	δ in F_13_	Δδ
α-(D)-glucose	92.94	92.71	0.23
β-(D)-glucose	97.51	97.35	0.16
α-(D)-fructofuranose	104.89	104.65	0.24
β-(D)-fructopyranose	102.65	102.47	0.18
β-(D)-fructopyranose	98.79	98.59	0.20

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
