# Peer review of "Dereplication of Natural Extracts Diluted in Glycerin: Physical Suppression of Glycerin by Centrifugal Partition Chromatography Combined with Presaturation of Solvent Signals in 13C-Nuclear Magnetic Resonance Spectroscopy"

_molecules, 2020, doi:10.3390/molecules25215061_

Round 1
Reviewer 1 Report
The paper by Canton M. and co-workers on the identification of natural extract from cosmetic mixture is really interesting from the industrial point of view. The possibility to understand the complex mixture of natural extract in glycerol is extremely important to understand the composition of cosmetic and pharmaceutical products especially for safety purpose.
The experimental part of the work is well described and the NMR analysis are clear and well explained also for people that are not inside this technique.
Minor revision:
In figure 4 page 8 the two NMR panels must be switched from (b)/ (a) to (a)/(b) to better understand the explanation.
Author Response
The authors thank reviewer 1 for her/his report.
As requested, the two horizontal panels in Figure 4 were switched.
Reviewer 2 Report
The authors studied combination of physical suppression by CPC (centrifugal partition chromatography) fractionation and spectroscopic suppression by presaturation in 13C NMR for the dereplication of natural extract metabolites diluted in glycerin. Principle of this approach was successfully achieved on a model mixture of 23 analytical standards. This method, applied on a bark extract of Cedrus atlantica Carrière diluted in glycerin, resulted in the identification of 12 metabolites of different chemical classes, therefore this approach efficient for the fast chemical profiling of natural extracts diluted in glycerin.
The article is written very clearly, the methods are appropriate, well described.
The article deserves to be published in its current form.
Author Response
The authors thank reviewer 2 for her/his report.